# Spatial Proximity of Cancer-Associated Fibroblasts to Tumor and Osteoclasts Suggests a Coordinating Role in OSCC-Induced Bone Invasion: A Preliminary Study

**DOI:** 10.3390/biomedicines13102554

**Published:** 2025-10-20

**Authors:** Nobuyuki Sasahara, Masayuki Kaneko, Takumi Kitaoka, Michihisa Kohno, Takanobu Kabasawa, Naing Ye Aung, Rintaro Ohe, Mitsuyoshi Iino, Mitsuru Futakuchi

**Affiliations:** 1Department of Pathology, Yamagata University Faculty of Medicine, Yamagata 9909585, Japan; 2Department of Dentistry, Oral and Maxillofacial·Plastic and Reconstructive Surgery, Yamagata University Faculty of Medicine, Yamagata 9909585, Japan; 3Department of Otolaryngology, Head and Neck Surgery, Yamagata University Faculty of Medicine, Yamagata 9909585, Japan; 4Department of Surgery I, Yamagata University Faculty of Medicine, Yamagata 9909585, Japan

**Keywords:** cancer-associated fibroblasts (CAFs), oral squamous cell carcinoma (OSCC), jawbone invasion, AI-assisted spatial histology, osteoclasts, tumor microenvironment

## Abstract

**Background**: Jawbone invasion is a common and prognostically unfavorable feature of oral squamous cell carcinoma (OSCC). Although cancer-associated fibroblasts (CAFs) are recognized for their role in tumor progression, their spatial dynamics at the tumor–bone interface remain poorly understood. **Methods**: We analyzed 14 OSCC specimens with confirmed jawbone invasion using histopathological and immunohistochemical techniques. Digital pathology combined with AI-assisted image analysis was employed to quantify and visualize the spatial distribution of OSCC cells (RANKL-positive), CAFs (α-SMA and FAP-positive), and osteoclasts (cathepsin K-positive) within defined regions of interest at the tumor–bone invasive front. **Results**: A consistent laminar stromal region enriched in CAFs was observed between the tumor nests and jawbone. CAFs were spatially clustered near OSCC cells and osteoclasts, with 81% and 74% residing within 50 μm, respectively. On average, 11.4 CAFs were present per OSCC cell and 23.2 per osteoclast. These spatial proximities were largely preserved irrespective of stromal thickness, suggesting active bidirectional cellular interactions. **Conclusions**: Our findings demonstrate that CAFs are strategically positioned to facilitate intercellular signaling between tumor cells and osteoclasts, potentially coordinating OSCC proliferation and bone resorption. This study highlights the utility of AI-assisted spatial histology in unraveling tumor microenvironmental dynamics and proposes CAFs as potential therapeutic targets in OSCC-induced osteolytic invasion.

## 1. Introduction

Oral squamous cell carcinoma (OSCC) is the most common histological subtype of head and neck cancers, accounting for more than 90% of all oral malignancies [1,2]. A hallmark of advanced OSCC—clinically and pathologically significant—is its strong propensity to invade adjacent jawbone structures, resulting in to osteolytic lesions [3,4]. Epidemiological studies report mandibular or maxillary invasion in approximately 12–56% of OSCC cases, a condition widely recognized as an adverse prognostic factor, strongly correlated with advanced staging and decreased survival rates [5,6].

The current standard of care for OSCC with jawbone invasion typically involves surgical resection of the affected bone, with the extent of resection determined by the degree of tumor infiltration [7]. However, such procedures are often associated with significant morbidity, including impaired mastication, dysarthria, and facial disfigurement, which can profoundly impact postoperative quality of life [8]. Therefore, a deeper understanding of the molecular and cellular mechanisms underlying jawbone invasion is essential for developing targeted therapeutic strategies aimed at reducing bone destruction while maintaining oncological control.

In the context of skeletal metastasis, bone resorption is primarily mediated by osteoclasts, rather than by direct degradation of mineralized tissue by tumor cells [9]. The differentiation and activation of osteoclasts are primarily regulated through the interaction between receptor activator of nuclear factor-κB ligand (RANKL) and its receptor RANK, which is expressed on osteoclast precursors. Osteoclastic bone resorption releases growth factors—such as transforming growth factor-β (TGF-β)—from the bone matrix, which in turn enhance tumor proliferation, thereby perpetuating the so-called “vicious cycle” of bone destruction and cancer progression [10].

A similar mechanism is believed to operate in OSCC-induced jawbone invasion, where tumor cell proliferation and osteoclast activation work synergistically to accelerate bone destruction [11]. Increasing attention has been directed toward the tumor microenvironment, especially the role of cancer-associated fibroblasts (CAFs), as key facilitators of tumor progression and stromal remodeling. CAFs are known to secrete a wide range of paracrine factors that promote OSCC cell proliferation, invasion, and therapy resistance [12]. Notably, emerging evidence suggests that conditioned media derived from CAFs can enhance osteoclast differentiation and activity in vitro, implicating them as potential contributors to tumor-induced osteolysis [3].

Based on these observations, we hypothesized that CAFs localized at the tumor–bone interface may orchestrate functional interactions between OSCC cells and osteoclasts, thereby facilitating jawbone invasion. To test this hypothesis, we established a novel in vivo platform capable of evaluating both direct and indirect interactions among CAFs, OSCC cells, and osteoclast precursors through AI-assisted spatial histological analysis. Using this system, we investigated the mechanistic role of CAFs in OSCC-induced bone invasion with the ultimate goal of identifying novel therapeutic targets to prevent osteolytic progression.

## 2. Materials and Methods

### 2.1. Patients and Tissue Specimens

Fourteen tissue specimens were collected from 14 patients diagnosed with oral squamous cell carcinoma (OSCC) exhibiting jawbone invasion at Yamagata University Hospital between August 2021 and April 2023.

Inclusion criteria were OSCC patients with histologically confirmed jawbone invasion who underwent primary resection at Yamagata University Hospital between August 2021 and April 2023. Exclusion criteria included prior neoadjuvant chemotherapy, recurrent disease, or insufficient bone-containing specimens. Primary tumor sites included gingiva (*n* = 8), tongue (*n* = 3), floor of mouth (*n* = 2), and buccal mucosa (*n* = 1). TNM stage was recorded for all patients. Since all cases were classified as T4a due to bone invasion, an association between TNM stage and bone invasion could not be assessed in this study.

Resected specimens were fixed in 10% neutral-buffered formalin at room temperature and subsequently embedded in paraffin. Specimens containing jawbone were decalcified using ethylenediaminetetraacetic acid (EDTA) prior to embedding. Tissue sections approximately 3 μm thick were prepared for hematoxylin and eosin (HE) staining and immunohistochemistry (IHC).

This study was approved by the Research Ethics Committee of the Faculty of Medicine, Yamagata University (approval no. 2023-53) and conducted in accordance with the principles outlined in the Declaration of Helsinki.

### 2.2. Histological Observation

All 14 OSCC specimens with confirmed jawbone invasion were examined histologically using hematoxylin and eosin (HE)-stained sections.

### 2.3. Immunohistochemical Analysis

Immunohistochemistry (IHC) was performed on three representative OSCC specimens with jawbone invasion.

All procedures were carried out using the BOND RXm autostainer (Leica Biosystems, Nussloch, Germany) in accordance with the manufacturer’s protocols. Immunoreactivity was visualized using brown staining with 3,3′-diaminobenzidine (DAB; BOND Polymer Refine Detection, Leica Biosystems), and sections were counterstained with hematoxylin (BOND Polymer Refine Detection, Leica Biosystems).

The following markers were analyzed: α-SMA (cytoplasmic; clone asm-1, Leica Biosystems), Fibroblast activation protein FAP (cytoplasmic/membranous; clone SP325, Abcam, Cambridge, UK), Receptor activator of nuclear factor-κB ligand, RANKL (membranous, polyclonal antibody, Abcam), and cathepsin K (cytoplasmic, polyclonal antibody, Abcam). Both internal (stromal fibroblasts, osteoclast precursors) and external controls (tonsil tissue) were used for validation. For serial sections (RANKL/FAP, FAP/cathepsin K), analyses were performed on the same cases (n = 3). Low-magnification observations were carried out on both HE- and IHC-stained sections as appropriate.

### 2.4. Quantification of OSCC Cells, Cancer-Associated Fibroblasts (CAFs), and Osteoclasts at the Invasive Front of the Jawbone

All tissue slides were digitized using the MoticEasyScan system (Motic Digital Pathology, San Francisco, CA, USA), and AI-assisted histological image analysis was conducted using HALO software (Indica Labs, Corrales, NM, USA). DAB-positive cells—including α-SMA-, FAP-, RANKL-, and cathepsin K-positive cells—were automatically detected using HALO.

To quantify CAFs in proximity to OSCC cells and osteoclasts at the tumor–bone invasive front, 30 regions of interest (ROIs) were selected per specimen. Each ROI measured 100 × 100 μm^2^. Fifteen ROIs were located along the bone surface (encompassing CAFs and osteoclasts), while the remaining fifteen were situated within deeper tumor nests (containing CAFs and OSCC cells). The number of each cell type was counted within each ROI, and the average number of CAFs per OSCC cell and per osteoclast was calculated using HALO. To ensure reproducibility, HALO detection thresholds for DAB-positive cells were uniformly applied across all samples and calibrated using control tissues. Cell size and signal intensity thresholds were held constant. ROI selection was independently performed by two observers in a blinded manner, and concordance was confirmed at >95%.

### 2.5. Analysis of Spatial Relationships Between OSCC Cells, CAFs, and Osteoclasts

To assess the spatial proximity between CAFs and OSCC cells, and between CAFs and osteoclasts, AI-assisted spatial histological analysis was conducted using HALO software. Serial tissue sections were stained for RANKL and FAP, or for FAP and cathepsin K. In the HALO analysis system, RANKL-positive OSCC cells, FAP-positive CAFs, and cathepsin K-positive osteoclasts were visualized as green dots, red rectangles, and blue triangles, respectively. These cellular markers were plotted on a coordinate grid, and the shortest distance from each OSCC cell or osteoclast to the nearest CAF was measured.

CAF distribution relative to OSCC cells and osteoclasts was analyzed in 10 μm radial increments. The number of CAFs within each distance range was counted, and histograms were generated to illustrate spatial distribution profiles.

## 3. Results

### 3.1. Induction of Cancer-Associated Fibroblasts (CAFs) Between the OSCC Growth Area and the Jawbone

At low magnification, OSCC cells were observed proliferating adjacent to stromal tissue (Figure 1A). Osteolytic changes attributed to osteoclast activity were evident on the surface of the jawbone. Notably, a distinct laminar stromal tissue layer was consistently interposed between the OSCC growth region and the jawbone.

High-magnification analysis of this stromal layer revealed that approximately 70% of the constituent cells exhibited a spindle-shaped morphology consistent with fibroblasts, while the remaining ~30% consisted of non-spindle-shaped cells, such as lymphocytes (Figure 1B). Immunohistochemical (IHC) staining confirmed that the spindle-shaped cells were positive for both α-smooth muscle actin (α-SMA) and fibroblast activation protein (FAP) (Figure 1C,D), confirming their identity as CAFs. These findings indicate a prominent presence of CAFs within the laminar stromal tissue positioned between the OSCC tumor mass and the jawbone.

### 3.2. Quantification of CAFs in the Vicinity of OSCC Cells and Osteoclasts

We hypothesized that CAFs preferentially localize near OSCC cells and osteoclasts due to specific cellular interactions. To evaluate this, we quantified OSCC cells, CAFs, and osteoclasts at the invasive front of the jawbone (Figure 2A).

IHC analysis revealed that receptor activator of nuclear factor-κB ligand (RANKL) was expressed exclusively in OSCC cells, and not in CAFs or osteoclasts (Figure 2B). HALO image analysis visualized RANKL-positive OSCC cells in green (Figure 2C), FAP-positive CAFs in red (Figure 2D,E), and cathepsin K-positive osteoclasts in blue (Figure 2F,G).

In randomly selected 100 × 100 μm^2^ regions containing OSCC cells and CAFs, 78 OSCC cells and 890 CAFs were detected, yielding a mean of 11.4 CAFs per OSCC cell. In areas containing osteoclasts and CAFs, 32 osteoclasts and 745 CAFs were observed, corresponding to an average of 23.2 CAFs per osteoclast.

### 3.3. Spatial Distribution of CAFs Relative to OSCC Cells

To further investigate the spatial association between CAFs and OSCC cells, we analyzed their distribution using HALO software. OSCC cells, visualized as green dots following RANKL staining (Figure 3A–C), and FAP-positive CAFs, visualized as red squares (Figure 3D–F), were plotted onto a spatial grid.

By superimposing these datasets, the shortest distance from each CAF to the nearest OSCC cell was calculated (Figure 3G). The number of CAFs was highest within 10 μm of OSCC cells and gradually declined with increasing distance (Figure 3H). Notably, 81% of CAFs located within 100 μm of OSCC cells were found within 50 μm, indicating that CAFs preferentially accumulate in close proximity to tumor cells.

### 3.4. Spatial Distribution of CAFs Relative to Osteoclasts

A similar analytical approach was applied to evaluate the spatial distribution of CAFs in relation to osteoclasts. FAP-positive CAFs (Figure 4A–C) and cathepsin K-positive osteoclasts (Figure 4D–F) were plotted on coordinate maps.

Following dataset superimposition (Figure 4G), the shortest distance from each CAF to the nearest osteoclast was measured. The histogram showed that 74% of CAFs located within 100 μm of osteoclasts were situated within 50 μm (Figure 4H), suggesting a preferential spatial proximity of CAFs to osteoclasts as well.

### 3.5. CAF Distribution and Stromal Thickness

Finally, we examined whether the vertical thickness of the laminar stromal tissue influenced the distribution of CAFs. In regions with thinner stromal tissue (Figure 5A), 89% of CAFs were located within 50 μm of OSCC cells, and 87% within 50 μm of osteoclasts (Figure 5B).

In contrast, in areas with thicker stromal tissue (Figure 5C), the proportion of CAFs located within 50 μm decreased to 62% for OSCC cells and 72% for osteoclasts (Figure 5D). These findings suggest that although CAFs preferentially localize near both OSCC cells and osteoclasts, the degree of proximity is influenced by stromal thickness.

In summary, across all 14 specimens, CAFs were consistently enriched at the tumor–bone interface, preferentially clustering near both OSCC cells and osteoclasts, supporting their hypothesized coordinating role.

## 4. Discussion

In this study, we identified a distinct laminar stromal layer at the tumor–bone interface in oral squamous cell carcinoma (OSCC). This observation is consistent with previous reports describing stromal cell induction at the tumor–bone interface in OSCC cases [3,13,14].

Notably, this phenomenon appears to differ from metastatic lesions of other cancers, such as breast and prostate cancer, in which stromal cellularity is typically sparse [15,16,17]. These findings suggest that a prominent stromal response is a distinguishing feature of OSCC-associated bone invasion.

Histological examination revealed that the stromal tissue is primarily composed of spindle-shaped cells (~70%), consistent with fibroblasts, and non-spindle-shaped cells (~30%), including lymphocytes. Immunohistochemical analysis confirmed that the majority of spindle-shaped cells express fibroblast activation protein (FAP) and α-smooth muscle actin (α-SMA), identifying them as cancer-associated fibroblasts (CAFs). Their consistent localization adjacent to both OSCC cells and jawbone osteoclasts suggests a dual role in modulating tumor proliferation and bone resorption.

CAFs are known to promote tumor growth and invasion through paracrine and juxtacrine mechanisms [18]. Spatial mapping in our study revealed that, on average, 11 CAFs were located in close proximity to each OSCC cell, supporting the possibility of direct CAF–tumor interactions. Likewise, approximately 23 CAFs were observed around each osteoclast, suggesting their potential involvement in osteoclastogenesis.

Because paracrine signaling is highly distance-dependent—soluble factor activity typically decreases in an exponential manner as the distance between source and target cells increases [19]—the spatial clustering of CAFs in close proximity to OSCC cells and osteoclasts is likely to provide a microenvironment in which intercellular communication can occur with maximal efficiency. This spatial arrangement may be particularly important in the bone–tumor interface, where rapid and localized signaling could amplify tumor-driven osteolysis [20]. Indeed, in the context of OSCC, CAFs have been associated with bone invasion and osteoclast activation, as demonstrated in a study showing CAFs from OSCC tissues enhanced osteoclastogenesis in vitro and correlated with bone resorption in vivo [3].

CAFs are known to secrete a broad spectrum of cytokines, chemokines, and growth factors that collectively contribute to tumor progression, angiogenesis, and modulation of immune responses [21,22,23]. In OSCC specifically, CAFs can influence tumor cell behavior via secreted factors such as TIAM1, which promotes proliferation, migration, invasion and epithelial–mesenchymal transition (EMT) in OSCC cells [24]. Moreover, CAFs in OSCC have been implicated in remodeling of the extracellular matrix via increased expression of lysyl oxidase (LOX), thus stiffening the matrix and promoting invasion via FAK signaling pathways [25]. Taken together, these findings strengthen the view that CAFs play a multifaceted role in OSCC progression and bone destruction.

Nevertheless, it is critical to recognize that CAFs do not represent a uniform entity but rather comprise a heterogeneous population encompassing multiple phenotypically and functionally distinct subtypes. In our study, CAFs were operationally defined on the basis of dual positivity for α-smooth muscle actin (α-SMA) and fibroblast activation protein (FAP).

While this combination provides a robust marker set for identifying activated fibroblasts, it may not capture the full spectrum of CAF diversity. Subpopulations such as α-SMA^+^/FAP^−^ or α-SMA^−^/FAP^+^ fibroblasts may be overlooked, despite their potential to exert unique effects on tumor progression, immune modulation, extracellular matrix remodeling, or osteoclastogenesis [21,26].

In OSCC, CAF heterogeneity has begun to be charted by single-cell RNA sequencing: one recent study compared fibroblasts in primary OSCC tumors and paired lymph nodes, revealing distinct phenotypic clusters and suggesting that local environment may drive CAF specialization [27]. Moreover, CAFs with particular markers may show enriched localization in bone-adjacent stroma in OSCC. For example, thymidine phosphorylase-positive CAFs (TP^+^ CAFs) were preferentially located at the surface of resorbed bone tissue in bone-invasive OSCC cases, and higher TP^+^ CAF density correlated with more aggressive bone invasion and poorer prognosis [28].

The functional contribution of CAF subsets could thus be context-dependent, varying according to tumor stage, microenvironmental cues, or interactions with immune and bone-resorbing cells [22,29]. To resolve this complexity and achieve a more comprehensive understanding, future investigations employing single-cell transcriptomic profiling, multiplexed imaging, and spatially resolved proteomic approaches will be essential [30,31,32]. Such methodologies would allow for precise delineation of CAF heterogeneity and for clarification of how individual subsets contribute to the intricate signaling networks driving OSCC invasion and bone resorption. Further studies should involve single-cell transcriptomic or multiplexed imaging approaches will be necessary to elucidate this complexity. Interestingly, although the vertical thickness of the stromal layer varied between 100 and 200 μm among specimens, CAFs consistently localized in close proximity to both OSCC cells and osteoclasts. This observation suggests that CAF positioning is independent of stromal thickness and may be directed by specific chemotactic gradients or mechanical cues.

Osteoclast differentiation is primarily mediated through receptor activator of nuclear factor-κB ligand (RANKL) signaling [10]. RANKL exists in both membrane-bound and soluble forms [33,34]. In our study, RANKL expression was detected in OSCC cells but not in CAFs, suggesting that tumor-derived RANKL is the main driver of osteoclastogenesis. However, it is plausible that CAFs further augment this process by secreting osteoclastogenic factors that synergize with tumor-derived RANKL.

To illustrate the versatility of AI-assisted histological spatial analysis, we briefly note its successful application in other pathological contexts (e.g., immune–stromal interactions in inflammatory disease and solid tumors) [35,36,37]. These examples highlight the broader potential of the technique while underscoring its specific relevance in OSCC.

These examples underscore the versatility of AI-assisted spatial histological analysis in revealing cellular architecture and signaling interactions across diverse pathological contexts. Applied to OSCC, this technique demonstrated that CAFs cluster densely around both tumor cells and osteoclasts, providing in vivo evidence of potential bidirectional interactions and suggesting a coordinating role for CAFs in tumor progression and bone resorption.

Although previous in vitro and ex vivo studies in OSCC have shown that cancer-associated fibroblasts (CAFs) can enhance osteoclastogenesis and are closely associated with bone invasion [3], our current findings do not directly establish their functional role in osteoclast differentiation or tumor invasion in vivo. Thus, despite the spatial proximity between CAFs, tumor cells, and osteoclasts suggesting a modulatory role at the invasive front, causality cannot be concluded from our dataset alone [38]. Accordingly, our interpretation remains hypothesis-generating rather than definitive.

Future studies should incorporate functional validation, including lineage or fate-mapping approaches to clarify CAF origin and persistence within OSCC lesions, systematic cytokine/chemokine profiling to identify mediators of CAF–osteoclast crosstalk (e.g., RANKL/OPG axis), and targeted perturbation of CAF–tumor or CAF–osteoclast signaling to test necessity and sufficiency for bone destruction [3,11]. Such experiments would enable a mechanistic dissection of how CAFs contribute to OSCC-associated osteolysis and progression. Nevertheless, our spatial histological analysis adds weight to a therapeutic rationale: CAFs accumulate at the tumor–bone interface and align with osteoclastogenic signaling in OSCC [3,11]. Together with reports linking specific CAF phenotypes (e.g., podoplanin-positive or HAS1^high^ subsets) to invasive behavior [39,40] and emerging CAF-focused diagnostics/theranostics (e.g., FAP-targeted imaging) in OSCC [41], these observations support consideration of CAFs as actionable targets in OSCC-associated bone destruction.

Beyond general cytokine signaling, candidate pathways may include the IL-6/STAT3 axis and TGF-β signaling, both of which are implicated in tumor–stroma–osteoclast crosstalk in related cancer models.

Future studies involving functional validation—such as lineage tracing, cytokine profiling, and targeted disruption of CAF–tumor or CAF–osteoclast signaling pathways—will be essential to definitively establish the mechanistic contributions of CAFs. Taken together, our AI-assisted spatial histological analysis supports the initial hypothesis that CAFs coordinate interactions between tumor cells and osteoclasts. The clinical implication is that targeting CAF-related signaling may help limit osteolytic invasion. Strengths of this study include the integration of AI-assisted spatial mapping with histopathological validation. Limitations include the relatively small sample size and descriptive nature of the analysis. Future work should involve functional validation, larger cohorts, and translational studies to establish therapeutic relevance.

## 5. Conclusions

We identified a distinct laminar stromal layer at the OSCC–bone interface, enriched with cancer-associated fibroblasts (CAFs) consistently localized near both tumor cells and osteoclasts. AI-assisted spatial analysis revealed close proximity conducive to paracrine signaling, supporting the hypothesis that CAFs may mediate intercellular crosstalk that promotes both tumor progression and bone destruction. The application of AI-assisted histological spatial analysis further enabled precise mapping of these cellular interactions in OSCC and other pathological conditions. Collectively, our findings highlight the spatially localized role of CAFs at the invasive tumor front and underscore their potential as therapeutic targets in OSCC-associated bone invasion. Targeting CAF-associated signaling pathways may represent a novel strategy to prevent tumor-induced osteolysis in OSCC.

## Figures and Tables

**Figure 1 biomedicines-13-02554-f001:**
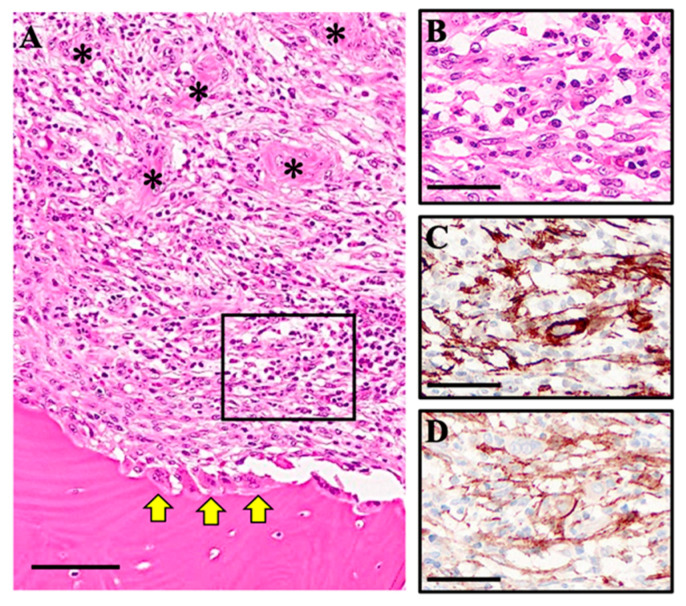
**Histology and representative immunohistochemistry at the OSCC–bone interface.** (**A**) H&E staining shows invasive OSCC nests (asterisks) extending toward mandibular bone; yellow arrows indicate the bone surface at the tumor–bone interface. The boxed region is magnified in (**B**), highlighting spindle-shaped stromal cells adjacent to the invasive front. (**C**,**D**) Immunohistochemistry (IHC) reveals abundant CAFs around the invasive front, detected by α-SMA (**C**) and FAP (**D**). Scale bars: 100 μm (**A**); 50 μm (**B**–**D**).

**Figure 2 biomedicines-13-02554-f002:**
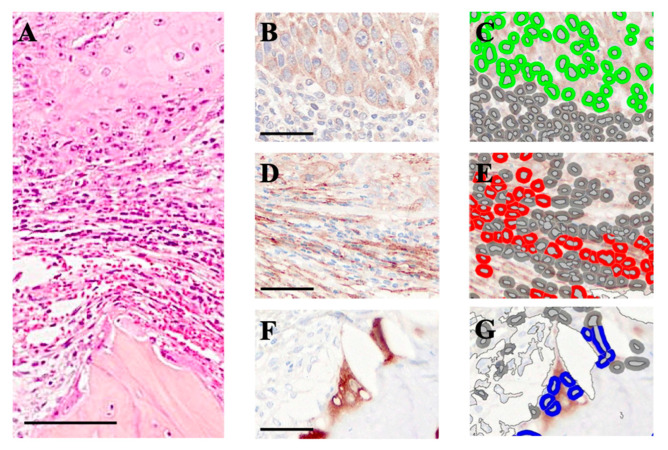
**Ground truth identification and digital annotation used for spatial analysis.** (**A**) H&E overview of the tumor–stroma–bone continuum at the invasive front. (**B**,**D**,**F**) IHC for RANKL ((**B**), OSCC cells), α-SMA ((**D**), CAFs), and cathepsin K ((**F**), osteoclasts). (**C**,**E**,**G**) Corresponding digital annotation maps: OSCC cells (green), CAFs (red), osteoclasts (blue). These masks provided coordinate data for subsequent AI-assisted spatial analyses. Scale bars: 100 μm (**A**); 50 μm (**B**–**G**).

**Figure 3 biomedicines-13-02554-f003:**
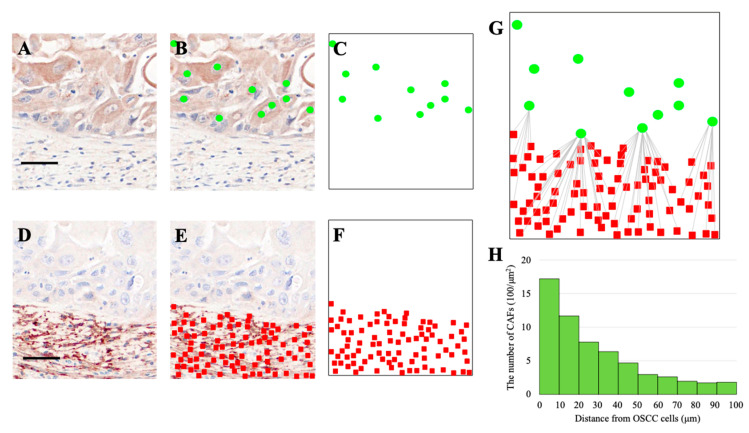
**Spatial proximity of CAFs to OSCC cells at the invasive front.** (**A**,**D**) IHC fields stained for RANKL (OSCC cells) and α-SMA (CAFs). (**B**,**C**) OSCC cells annotated as green circles. (**E**,**F**) CAF positions annotated as red squares. (**G**) Nearest-neighbor diagram: each CAF (red square) is connected to its closest OSCC cell (green circle) by a thin gray line. (**H**) Histogram showing CAF density (per 100 μm^2^) plotted as a function of distance from OSCC cells (10 μm bins, 0–100 μm). Scale bars: 50 μm.

**Figure 4 biomedicines-13-02554-f004:**
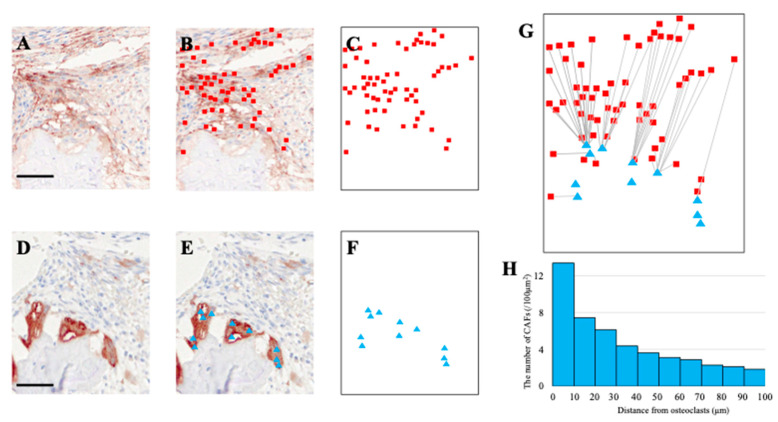
**Spatial proximity of CAFs to osteoclasts at the tumor–bone interface.** (**A**,**D**) IHC fields stained for α-SMA (CAFs) and cathepsin K (osteoclasts). (**B**) CAF positions annotated as red squares. (**C**) CAF distribution map used for quantification. (**E**,**F**) Osteoclast positions annotated as blue triangles. (**G**) Nearest-neighbor diagram: each CAF (red square) is linked to its closest osteoclast (blue triangle). (**H**) Histogram showing CAF density (per 100 μm^2^) relative to distance from osteoclasts (10 μm bins, 0–100 μm). Scale bars: 50 μm.

**Figure 5 biomedicines-13-02554-f005:**
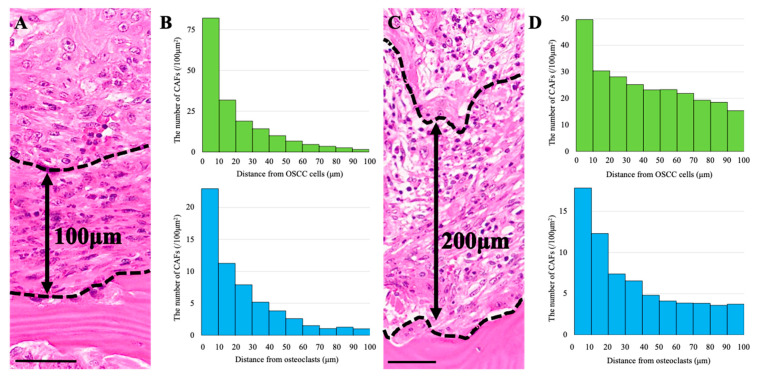
**CAF gradients with respect to OSCC cells and osteoclasts in shallow versus deep invasion.** (**A**,**C**) H&E sections of OSCC cases with shallow ((**A**), invasion depth ≈ 100 μm) or deep ((**C**), ≈200 μm) bone invasion; dotted lines indicate the tumor–bone interface. (**B**,**D**) Histograms show CAF density gradients (per 100 μm^2^) relative to OSCC cells (upper panels, green) and osteoclasts (lower panels, blue), calculated in 10 μm bins up to 100 μm. Scale bars: 50 μm.

## Data Availability

The data are not publicly available due to privacy or ethical restrictions.

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
