# Peer review of "Spatial Proximity of Cancer-Associated Fibroblasts to Tumor and Osteoclasts Suggests a Coordinating Role in OSCC-Induced Bone Invasion: A Preliminary Study"

_biomedicines, 2025, doi:10.3390/biomedicines13102554_

Round 1

Reviewer 1 Report

Comments and Suggestions for Authors

hello

thank you very much for a very interesting paper

The OSCC infiltrating the bone is quite challenging in all cases, since the scope of DOI and associated bone features are sometimes hard to identify and choose  a good approach, rather than a total radical resection

title is very interesting

title matches the abstract

abstract is well organised and structured with clear aim

used key words aresufficient and good

abstract and title are sound, nothing to change

introduction - its very nicely presented

introduction is presenting most imporant remarks in a good manner, nothing to change

introduction has a clear aim and is supported with a detailed hypothesis

authors did a great work in the introduction

there is no need to improve the introduction

The material and methods section is divided into chapters and divisions - this format is very good, because it enables the paper to be clear and written in a step by step manner

Methodology and material is well described

perhaps the 14 sample is a very small amount, and therefore I would recomment authors to change the title to preliminary study on.....

because 14 its not yet a big number, and Im quite sure that the authors will still continue those studies on OSCC bone invasion in the future

ethic standards are supported

despite an information on 14 patients - I dont really see a full inlusion and exclusion criteria for the study? perhaps write them or add a table? also, did TNM influence on bone invasion and DOI in histopathology?this needs to be clarified and explained in text

was bone invasion caused only by gingival cancer? or what were the primary location of the cancers? please add this information

rest of authors methodology is very nicely presented

results are good and very nicely written

figure 1 is very nicely presnted, nothing more to add or change

the results are well presented in sub chapters with figure and a clear explanation

what kind of information does the reuslt present for a typical surgeon?

chapters 3.1-3.5 and figures 1-5 are very interesting and they are all connected and linked togeter with a clear message and support authors' results and aim of the study

discussion is short - please improve it and write how exactly the scope of bone invasion can help in daily oncologic center work?

used references and topics in the discussion are good, nothing more to change

im misisng some study limitations, that should be presented as the last chapter at the end of discussion - please improve it

ocerall scientific merit and paper results and conclusions are very interesting

if the paper is improved based on my comments, It would become a very good paper and a possible source for future citations

references are good and solid, nothing more to change

presented paper is very challenging and should be considered for further proceedings, when some minor changes are made

thank you very much for this intersting paper

authors did a very good job writing this paper, thank you and congratluations

Author Response

Reviewer 1

Comment 1: The sample size (n=14) is small; consider changing the title to indicate this is a preliminary study.

Response: We have revised the title to:

Spatial Proximity of Cancer-Associated Fibroblasts to Tumor and Osteoclasts Suggests a Coordinating Role in OSCC-Induced Bone Invasion: A Preliminary Study (line 2–3).

Comment 2: Inclusion/exclusion criteria are not clearly described. Please add these details.

Response: We added explicit inclusion and exclusion criteria in the Methods section as follow; Inclusion criteria were OSCC patients with histologically confirmed jawbone invasion who underwent primary resection at Yamagata University Hospital between August 2021 and April 2023. Exclusion criteria included prior neoadjuvant chemotherapy, recurrent disease, or insufficient bone-containing specimens. (Line 102-109)

Comment 3: Did TNM influence bone invasion and DOI? What were the primary locations of the cancers? Please add this information.

Response: We included primary tumor sites (gingiva, tongue, floor of mouth, buccal mucosa) and TNM stage information in the Methods section as follows.

Primary tumor sites included gingiva (n=8), tongue (n=3), floor of mouth (n=2), and buccal mucosa (n=1). TNM stage was recorded for all patients. Since all cases were classified as T4a due to bone invasion, an association between TNM stage and bone invasion could not be assessed in this study.(Line 105-109)

Comment 4: The Discussion is short; please expand on the clinical implications of bone invasion and state study limitations.

Response: We expanded the Discussion to support our findings.

We added the new paragraph in the discussion section

Although previous in vitro and ex vivo studies in OSCC have shown that cancer-associated fibroblasts (CAFs) can enhance osteoclastogenesis and are closely associated with bone invasion [3], our current findings do not directly establish their functional role in osteoclast differentiation or tumor invasion **in vivo**. Thus, despite the spatial proximity between CAFs, tumor cells, and osteoclasts suggesting a modulatory role at the invasive front, causality cannot be concluded from our dataset alone [38]. Accordingly, our interpretation remains hypothesis-generating rather than definitive.

Future studies should incorporate functional validation, including lineage or fate-mapping approaches to clarify CAF origin and persistence within OSCC lesions, systematic cytokine/chemokine profiling to identify mediators of CAF–osteoclast crosstalk (e.g., RANKL/OPG axis), and targeted perturbation of CAF–tumor or CAF–osteoclast signaling to test necessity and sufficiency for bone destruction [3,11]. Such experiments would enable a mechanistic dissection of how CAFs contribute to OSCC-associated osteolysis and progression. Nevertheless, our spatial histological analysis adds weight to a therapeutic rationale: CAFs accumulate at the tumor–bone interface and align with osteoclastogenic signaling in OSCC [3,11]. Together with reports linking specific CAF phenotypes (e.g., podoplanin-positive or HAS1high subsets) to invasive behavior [39,40] and emerging CAF-focused diagnostics / theranostics (e.g., FAP-targeted imaging) in OSCC [41], these observations support consideration of CAFs as actionable targets in OSCC-associated bone destruction.(Line 333-353)

We added the following paragraph to highlight clinical implications for surgical decision-making, study limitations, and future directions.

Taken together, our AI-associated spatial histological analysis supports the initial hypothesis that CAFs coordinate interactions between tumor cells and osteoclasts. The clinical implication is that targeting CAF-related signaling may help limit osteolytic invasion. Strengths of this study include the integration of AI-assisted spatial mapping with histopathological validation. Limitations include the relatively small sample size and descriptive nature of the analysis. Future work should involve functional validation, larger cohorts, and translational studies to establish therapeutic relevance.(Line360-367

Reviewer 2 Report

Comments and Suggestions for Authors

This is a well-conducted and clearly presented study that uses AI-assisted spatial histology to provide compelling in vivo evidence for the strategic positioning of Cancer-Associated Fibroblasts (CAFs) at the invasive front of oral squamous cell carcinoma (OSCC) with jawbone invasion. The manuscript is well-written, and the figures are of high quality. The finding that CAFs are consistently located in close proximity to both OSCC cells and osteoclasts provides a strong anatomical basis for the hypothesis that they play a coordinating role in tumor progression and bone resorption. I recommend this manuscript for publication after minor revisions.

My suggestions are as follows:

1. Clarification of Sample Size for IHC and Spatial Analysis: In section 2.1, it is stated that 14 tissue specimens were collected. However, section 2.3 states that immunohistochemistry (IHC) was performed on "three representative OSCC specimens". It is unclear if the subsequent detailed quantitative and spatial analyses (sections 2.4, 2.5, and all results in section 3) were also based on only these three specimens or on the full cohort of 14. If the quantitative data is derived from only three patients, this is a significant limitation and should be explicitly stated in the manuscript to manage reader expectations. Please clarify in the methods section how many of the 14 specimens were subjected to the full AI-assisted HALO analysis.

2. Discussion of Other Disease Models: While the examples from IgG4-related sialadenitis, TNBC, and CRC effectively demonstrate the versatility of the AI-assisted spatial analysis technique , this section feels somewhat disconnected from the main narrative of OSCC. Consider slightly shortening this paragraph to focus more on the principle of the technique's utility rather than detailing the specific quantitative results from these other unrelated diseases. This would help maintain a tighter focus on the primary subject of the paper.

3. The authors correctly acknowledge that the study is descriptive and does not establish a functional role for CAFs. In the discussion, it might strengthen the paper to briefly speculate on which specific signaling pathways (e.g., IL-6/STAT3, TGF-β, etc.), beyond general "cytokines and growth factors," might be the most promising candidates for mediating this tripartite interaction in OSCC, based on existing literature. This could provide a more concrete direction for future functional studies.

Overall, this is an excellent and valuable contribution to the field. The requested revisions are minor and intended to improve the clarity and impact of the manuscript.

Author Response

Reviewer 2

Comment 1: Clarification of sample size for IHC and spatial analysis. Were the quantitative data derived from 3 or 14 cases?

Response: We clarified that IHC and serial section analyses were performed on three representative specimens, while quantitative and spatial analyses were conducted on the full cohort of 14 specimens

(Methods, lines 123-124).

Comment 2: The discussion of other disease models (IgG4-related sialadenitis, TNBC, CRC) feels disconnected; consider shortening.

Response: We shortened this section (Discussion, lines 271–285), focusing on the principle of the technique’s versatility rather than detailed quantitative data.

We revised as follow: To illustrate the versatility of AI-assisted histological spatial analysis, we briefly note its successful application in other pathological contexts (e.g., immune–stromal interactions in inflammatory disease and solid tumors) [35-37]. These examples highlight the broader potential of the technique while underscoring its specific relevance in OSCC.(Line322-326)

Comment 3: Consider briefly speculating on specific signaling pathways that may mediate the CAF–tumor–osteoclast interaction (e.g., IL-6/STAT3, TGF-β).

Response: We expanded the Discussion (lines 261-308, 333–356) to include IL-6/STAT3 and TGF-β as candidate pathway as follow

Line 261-308

Because paracrine signaling is highly distance-dependent—soluble factor activity typically decreases in an exponential manner as the distance between source and target cells increases [19]—the spatial clustering of CAFs in close proximity to OSCC cells and osteoclasts is likely to provide a microenvironment in which intercellular communication can occur with maximal efficiency. This spatial arrangement may be particularly important in the bone–tumor interface, where rapid and localized signaling could amplify tumor-driven osteolysis [20]. Indeed, in the context of OSCC, CAFs have been associated with bone invasion and osteoclast activation, as demonstrated in a study showing CAFs from OSCC tissues enhanced osteoclastogenesis in vitro and correlated with bone resorption in vivo [3].

CAFs are known to secrete a broad spectrum of cytokines, chemokines, and growth factors that collectively contribute to tumor progression, angiogenesis, and modulation of immune responses [21-23]. In OSCC specifically, CAFs can influence tumor cell behavior via secreted factors such as TIAM1, which promotes proliferation, migration, invasion and epithelial–mesenchymal transition (EMT) in OSCC cells [24]. Moreover, CAFs in OSCC have been implicated in remodeling of the extracellular matrix via increased expression of lysyl oxidase (LOX), thus stiffening the matrix and promoting invasion via FAK signaling pathways [25]. Taken together, these data strengthen the view that CAFs play a multifaceted role in OSCC progression and bone destruction.

Nevertheless, it is critical to recognize that CAFs do not represent a uniform entity but rather comprise a heterogeneous population encompassing multiple phenotypically and functionally distinct subtypes. In our study, CAFs were operationally defined on the basis of dual positivity for α-smooth muscle actin (α-SMA) and fibroblast activation protein (FAP).

While this combination provides a robust marker set for identifying activated fibroblasts, it may not capture the full spectrum of CAF diversity. Subpopulations such as α-SMA⁺/FAP⁻ or α-SMA⁻/FAP⁺ fibroblasts may be overlooked, despite their potential to exert unique effects on tumor progression, immune modulation, extracellular matrix remodeling, or osteoclastogenesis [21,26].

In OSCC, CAF heterogeneity has begun to be charted by single-cell RNA sequencing: one recent study compared fibroblasts in primary OSCC tumors and paired lymph nodes, revealing distinct phenotypic clusters and suggesting that local environment may drive CAF specialization [27]. Moreover, CAFs with particular markers may show enriched localization in bone-adjacent stroma in OSCC. For example, thymidine phosphorylase–positive CAFs (TP⁺ CAFs) were preferentially located at the surface of resorbed bone tissue in bone-invasive OSCC cases, and higher TP⁺ CAF density correlated with more aggressive bone invasion and poorer prognosis [28].

The functional contribution of CAF subsets could thus be context-dependent, varying according to tumor stage, microenvironmental cues, or interactions with immune and bone-resorbing cells [22,29].To resolve this complexity and achieve a more comprehensive understanding, future investigations employing single-cell transcriptomic profiling, multiplexed imaging, and spatially resolved proteomic approaches will be essential [30-32]. Such methodologies would allow for precise delineation of CAF heterogeneity and for clarification of how individual subsets contribute to the intricate signaling networks driving OSCC invasion and bone resorption.

Line 333-356

Although previous in vitro and ex vivo studies in OSCC have shown that cancer-associated fibroblasts (CAFs) can enhance osteoclastogenesis and are closely associated with bone invasion [3], our current findings do not directly establish their functional role in osteoclast differentiation or tumor invasion **in vivo**. Thus, despite the spatial proximity between CAFs, tumor cells, and osteoclasts suggesting a modulatory role at the invasive front, causality cannot be concluded from our dataset alone [38]. Accordingly, our interpretation remains hypothesis-generating rather than definitive.

Future studies should incorporate functional validation, including lineage or fate-mapping approaches to clarify CAF origin and persistence within OSCC lesions, systematic cytokine/chemokine profiling to identify mediators of CAF–osteoclast crosstalk (e.g., RANKL/OPG axis), and targeted perturbation of CAF–tumor or CAF–osteoclast signaling to test necessity and sufficiency for bone destruction [3,11]. Such experiments would enable a mechanistic dissection of how CAFs contribute to OSCC-associated osteolysis and progression. Nevertheless, our spatial histological analysis adds weight to a therapeutic rationale: CAFs accumulate at the tumor–bone interface and align with osteoclastogenic signaling in OSCC [3,11]. Together with reports linking specific CAF phenotypes (e.g., podoplanin-positive or HAS1high subsets) to invasive behavior [39,40] and emerging CAF-focused diagnostics / theranostics (e.g., FAP-targeted imaging) in OSCC [41], these observations support consideration of CAFs as actionable targets in OSCC-associated bone destruction.

Beyond general cytokine signaling, candidate pathways may include the IL-6/STAT3 axis and TGF-β signaling, both of which are implicated in tumor–stroma–osteoclast crosstalk in related cancer models.

Reviewer 3 Report

Comments and Suggestions for Authors

Dear Author,

Please find he comment attached 

Regards

Author Response

Reviewer 3

Comment 1: Please enumerate factors evaluated in the 14 specimens.
Response: We revised the Methods to specify the evaluated histological factors in all 14 OSCC specimens with jawbone invasion, as follow; Primary tumor sites included gingiva (n=8), tongue (n=3), floor of mouth (n=2), and buccal mucosa (n=1).(Line 105-109)

Comment 2: For the immunohistochemistry (IHC) performed on three representative cases, kindly specify the markers analyzed.

Response: We added these details in the Methods as follows

The following markers were analyzed: α-SMA (cytoplasmic ; clone asm-1, Leica Biosystems), Fibroblast activation protein FAP (cytoplasmic / membranous ; clone SP325, Abcam, Cambridge, UK), Receptor activator of nuclear factor-κB ligand, RANKL (membranous, polyclonal antibody, Abcam), and cathepsin K (cytoplasmic, polyclonal antibody, Abcam). Both internal (stromal fibroblasts, osteoclast precursors) and external controls (tonsil tissue) were used for validation. For serial sections (RANKL/FAP, FAP/cathepsin K), analyses were performed on the same cases (n=3).(Line 132-140)

Comment 3: Indicate the site of staining (nuclear, cytoplasmic, membranous).

Response: The following details were added in the Methods:
α-SMA (cytoplasmic ; clone asm-1, Leica Biosystems), Fibroblast activation protein FAP (cytoplasmic / membranous ; clone SP325, Abcam, Cambridge, UK), Receptor activator of nuclear factor-κB ligand, RANKL (membranous, polyclonal antibody, Abcam), and cathepsin K (cytoplasmic, polyclonal antibody, Abcam).(Line 131- 136)

Comment 4: Describe internal and external controls.
Response: We clarified in the Methods that both internal controls (stromal fibroblasts, osteoclast precursors) and external controls (tonsil tissue) were used for validation.as follow:

Both internal (stromal fibroblasts, osteoclast precursors) and external controls (tonsil tissue) were used for validation. For serial sections (RANKL/FAP, FAP/cathepsin K), analyses were performed on the same cases (n=3).(Line 136-139)

Comment 5: Clarify whether serial sections (RANKL/FAP, FAP/cathepsin K) were from the same cases, and how many.
Response: This information has been added as follows

For serial sections (RANKL/FAP, FAP/cathepsin K), analyses were performed on the same cases (n=3). Low-magnification observations were carried out on both HE- and IHC-stained sections as appropriate.(Line 137-140)

Comment 6: Specify whether low-magnification observations were based on HE or IHC sections.
Response: This has been clarified in Methods as follows

Low-magnification observations were carried out on both HE- and IHC-stained sections as appropriate.(Line 139-140)

Comment 7: Please add an overall summary at the end of the Results.
Response: We added a summary statement at the end of Results as follows

In summary, across all 14 specimens, CAFs were consistently enriched at the tumor–bone interface, preferentially clustering near both OSCC cells and osteoclasts, supporting their hypothesized coordinating role.(Line 234-236)

Comment 8: The discussion of CAF clustering should be supported with additional literature.
Response: We added the new paragraph in the discussion section to support our findings.

Although previous in vitro and ex vivo studies in OSCC have shown that cancer-associated fibroblasts (CAFs) can enhance osteoclastogenesis and are closely associated with bone invasion [3], our current findings do not directly establish their functional role in osteoclast differentiation or tumor invasion in vivo. Thus, despite the spatial proximity between CAFs, tumor cells, and osteoclasts suggesting a modulatory role at the invasive front, causality cannot be concluded from our dataset alone [38]. Accordingly, our interpretation remains hypothesis-generating rather than definitive.

Future studies should incorporate functional validation, including lineage or fate-mapping approaches to clarify CAF origin and persistence within OSCC lesions, systematic cytokine/chemokine profiling to identify mediators of CAF–osteoclast crosstalk (e.g., RANKL/OPG axis), and targeted perturbation of CAF–tumor or CAF–osteoclast signaling to test necessity and sufficiency for bone destruction [3,11]. Such experiments would enable a mechanistic dissection of how CAFs contribute to OSCC-associated osteolysis and progression. Nevertheless, our spatial histological analysis adds weight to a therapeutic rationale: CAFs accumulate at the tumor–bone interface and align with osteoclastogenic signaling in OSCC [3,11]. Together with reports linking specific CAF phenotypes (e.g., podoplanin-positive or HAS1high subsets) to invasive behavior [39,40] and emerging CAF-focused diagnostics / theranostics (e.g., FAP-targeted imaging) in OSCC [41], these observations support consideration of CAFs as actionable targets in OSCC-associated bone destruction. (Line 333-353)

Comment 9: The methods should explicitly emphasize that this was AI-assisted analysis.
Response: We revised the text throughout to emphasize AI-assisted histological spatial analysis (e.g., Lines 45, 49, 92, 163, 322, 359, 360, 363, 372, 396)

Comment 10: In the Discussion, explicitly state whether the hypothesis was satisfied, highlight clinical relevance, strengths, limitations, and future directions.

Response: We added these points at the end of the Discussion as follows:

Taken together, our AI-associated spatial histological analysis supports the initial hypothesis that CAFs coordinate interactions between tumor cells and osteoclasts. The clinical implication is that targeting CAF-related signaling may help limit osteolytic invasion. Strengths of this study include the integration of AI-assisted spatial mapping with histopathological validation. Limitations include the relatively small sample size and descriptive nature of the analysis. Future work should involve functional validation, larger cohorts, and translational studies to establish therapeutic relevance.(Line 360-367)

Round 2

Reviewer 1 Report

Comments and Suggestions for Authors

hello

thank you for all the changes

now [aper is improved

this preliminary study might add new insight into oscc

thank you

I strongly encourage authors to continue their great work in this scope

all the best for all authors

Reviewer 2 Report

Comments and Suggestions for Authors

I agree to this article being published in its current form, but please ask the author to carefully check for any minor errors, such as grammatical mistakes and typos.